# Experimental impacts of grazing on grassland biodiversity and function are explained by aridity

Minna Zhang[1], Manuel Delgado-Baquerizo [2,3], Guangyin Li[1,4], Forest Isbell [5], Yue Wang[1], Yann Hautier[6], Yao Wang[1], Yingli Xiao[1], Jinting Cai[1], Xiaobin Pan[1] & Ling Wang [1] ✉

Grazing by domestic herbivores is the most widespread land use on the planet, and also a major global change driver in grasslands. Yet, experimental evidence on the long-term impacts of livestock grazing on biodiversity and function is largely lacking. Here, we report results from a network of 10 experimental sites from paired grazed and ungrazed grasslands across an aridity gradient, including some of the largest remaining native grasslands on the planet. We show that aridity partly explains the responses of biodiversity and multifunctionality to long-term livestock grazing. Grazing greatly reduced biodiversity and multifunctionality in steppes with higher aridity, while had no effects in steppes with relatively lower aridity. Moreover, we found that long-term grazing further changed the capacity of above- and below-ground biodiversity to explain multifunctionality. Thus, while plant diversity was positively correlated with multifunctionality across grasslands with excluded livestock, soil biodiversity was positively correlated with multifunctionality across grazed grasslands. Together, our cross-site experiment reveals that the impacts of long-term grazing on biodiversity and function depend on aridity levels, with the more arid sites experiencing more negative impacts on biodiversity and ecosystem multifunctionality. We also highlight the fundamental importance of conserving soil biodiversity for protecting multifunctionality in widespread grazed grasslands.

Grasslands are among the most widespread and diverse ecosystems on Earth, covering >40% of terrestrial surface and supporting a wide range of biodiversity and ecosystem services for humankind[1]. In these ecosystems, grazing by livestock is a major driver of biodiversity and function[2–6], constituting the most widespread land use worldwide[7]. Grazing by livestock is essential for the production of food and for the proficiency of local economies, however, there are also major concerns about the sustainability of managed grazing, in part because of the large domestic livestock biomass in many managed grazing systems[8].

[1]Institute of Grassland Science, Key Laboratory of Vegetation Ecology of the Ministry of Education, Jilin Songnen Grassland Ecosystem National Observation and Research Station, Northeast Normal University, Changchun, China. [2]Laboratorio de Biodiversidad y Funcionamiento Ecosistémico. Instituto de Recursos Naturales y Agrobiología de Sevilla (IRNAS), CSIC, Sevilla, Spain. [3]Unidad Asociada CSIC-UPO (BioFun). Universidad Pablo de Olavide, Sevilla, Spain. [4]Key Laboratory of Wetland Ecology and Environment, Heilongjiang Xingkai Lake Wetland Ecosystem National Observation and Research Station, Northeast Institute of Geography and Agroecology, Chinese Academy of Sciences, Changchun, China. [5]Department of Ecology, Evolution and Behavior, University of Minnesota, Saint Paul, MN, USA. [6]Ecology and Biodiversity Group, Department of Biology, Utrecht University, Utrecht, the Netherlands. ✉e-mail: wangl890@nenu.edu.cn

Currently there are four major groups of uncertainties when studying grazing impacts on biodiversity and ecosystem functions. First, aridity is the most important feature in drylands, yet the experimental interaction between aridity and grazing impacts on biodiversity and multifunctionality have not been studied before. This limitation stems from the fact that most previous grazing experiments are conducted at a single site[9–11]. Regional experiments are needed to determine whether domestic herbivore impact on biodiversity and ecosystem functions is general or whether these effects depend on grassland aridity conditions. Second, current work exploring the interactions between climate and grazing on functions are based on meta-analyses[12–14] (i.e., unstandardized sampling and analytical approaches) and unstandardized field investigation (i.e., grazed and ungrazed grassland distributed in different sites)[15]. Yet, regional paired comparison and replicated site experiments investigating the impacts of grazing on biodiversity and multiple ecosystem functions are urgently needed to empirically determine the causal effects of grazing on biodiversity and multifunctionality. Third, most empirical examinations of grazing effects are based on short-term grazing duration[16–20], however, given there may be time-lag in the responses of ecosystem to grazing, long-term experiments to examine how livestock grazing can impact biodiversity and multifunctionality are needed to provide more solid evidence. Finally, while the contribution of biodiversity to supporting multiple ecosystem functions is well-established in natural ecosystems[21–26], recent studies have begun to reveal the relative importance of above-ground (plants) and below-ground (soil microbes) biodiversity for ecosystem functioning[27–29]. This knowledge is needed in order to improve the understanding of the relationships between biodiversity and ecosystem functioning thereby formulate sustainable conservation and management policies. However, it's still unclear whether and how long-term livestock grazing can impact the relative contribution of above and belowground biodiversity to ecosystem multifunctionality. Altogether, these uncertainties hamper our ability to better manage the environmental impacts of continued domestic herbivore grazing- the most extensive land use on the planet.

Here, we used a network of field experiments with paired grazing conditions (including and excluding livestock) carried out across 10 public rangelands experiencing decades of grazing (Supplementary Table 1, Supplementary Fig. 1) to investigate the impacts of long-term grazing on biodiversity and ecosystem multifunctionality. The exclusion of livestock was maintained over 10 years at each site, allowing us to get comparable ungrazed grasslands and thereby examine livestock grazing effects. The gradient in aridity includes the three major types of grasslands in northern China (meadow steppe, typical steppe, and desert steppe from wetter to drier)—the largest natural grasslands remaining on Earth. We aimed to examine the effects of domestic herbivore grazing on biodiversity and ecosystem multifunctionality (EMF), and whether the effects depend on grassland aridity, and whether long-term grazing change the relative strength of plant and soil biodiversity in supporting EMF. The EMF was associated with 11 functions (above-ground biomass, below-ground biomass, plant community N, plant community P, soil organic C, soil available N, microbial biomass C, microbial biomass N, decomposers, pathogen control, and mycorrhizal mutualism; see Methods). These surrogates of function constitute a good proxy for productivity, nutrient cycling, and build-up of nutrient pools, which are important determinants of ecosystem functioning in grazed grasslands. We determined multi-diversity, belowground biodiversity (soil microbial diversity), and averaging multifunctionality by averaging the standardized values of different organism groups and ecosystem functions, respectively (Methods). Ecosystem multifunctionality was additionally calculated considering multiple aspects of ecosystem functions, weighted multifunctionality (weighted EMF[30]), individual functions, and number of functions over a given functional threshold (multi-threshold EMF[31]).

To our knowledge, this is the first large-scale multi-site experiment examining the effect of long-term domestic herbivore grazing on grassland biodiversity and ecosystem multifunctionality across an aridity gradient. We hypothesized that long-term domestic herbivore grazing had stronger negative impacts on biodiversity and EMF in more arid sites because of the low net primary production and plant nutrient uptake characterizing these ecosystems. We further hypothesized that soil biodiversity may gain importance for supporting multifunctionality in long-term grazed ecosystems wherein plant communities may be more altered by the direct impact of grazers.

## Results and discussion

Our work provides empirical evidence that the long-term impact of domestic herbivore on biodiversity and multifunctionality are driven by aridity, with grazing being especially damaging for supporting functions in the most arid sites. Thus, our work advances current observational-level knowledge on how grazing interacts with climate to explain multiple ecosystem functions[15] by providing long-term evidence from a network of field experiments. We further provide evidence that decades of livestock grazing shifted the biodiversity drivers of multifunctionality from plant diversity to soil biodiversity when moving from ungrazed to grazed ecosystems, highlighting the relevance of soil biodiversity conservation in the most arid ecosystems.

We found that grazing interacted with grassland type (i.e., closely related with aridity) to determine biodiversity and multifunctionality, and the interactive effects for biodiversity was from Shannon's diversity index, which includes species richness and evenness (Supplementary Table 3). Specifically, long-term grazing had no effects in meadow steppes with relative lower aridity, but reduced biodiversity and multifunctionality in desert steppes with higher aridity (Figs. 1a and 2a). Moreover, the negative effect was gradually strengthened with aridity (Figs. 1b and 2b), thus grazing reduced multifunctionality more in desert steppes than in typical steppe (Fig. 1a). Desert steppes are fragile ecosystems, which generally suffer severe wind erosion for topsoil, especially when they experience external disturbances such as grazing[32, 33], which may contribute to the decline in ecosystem functions. Future studies also need to further examine long-term grazing-induced changes in subsoil functions. These results suggest that grasslands with high aridity may be more vulnerable to livestock grazing, compared with grasslands with lower aridity. These results also indicate that limiting grazing pressure through livestock removal or shortening of the duration of grazing is necessary in rangeland regions with aridity intensification as global climate change.

Both plant diversity and soil biodiversity have been recognized to be important drivers of ecosystem functions in grasslands[34–38]. We found that long-term grazing did not influence plant diversity of steppes with higher aridity (Fig. 2c). Instead, soil biodiversity was more sensitive to the synergistic negative effects of grazing and aridity, which was greatly decreased in the desert steppe with strong aridity (Fig. 2d), and thus may contribute to the decline in multifunctionality at arid grasslands. More importantly, our findings further provide evidence that a shift in the role from above- to below-ground diversity maintains grassland multifunctionality when moving from long-term domestic herbivore exclusion to grazing. That is, soil biodiversity supported multifunctionality under long-term livestock grazing, while plant diversity supported multifunctionality after over a decade of livestock exclusion (Supplementary Table 4; Fig. 3). Here, we provide the experimental evidence showing that long-term livestock grazing can shift the relative contribution of above and belowground biodiversity drivers in supporting multifunctionality (Fig. 3). These results were maintained for most of the individual functions measured, as well as the possible combinations among functions (Fig. S8), and also for the approach used to quantify multifunctionality (Supplementary Table 4, Supplementary Fig. 11 and 12): averaging multifunctionality (Fig. 3c, d), multiple-threshold multifunctionality (Fig. 3a, b), and

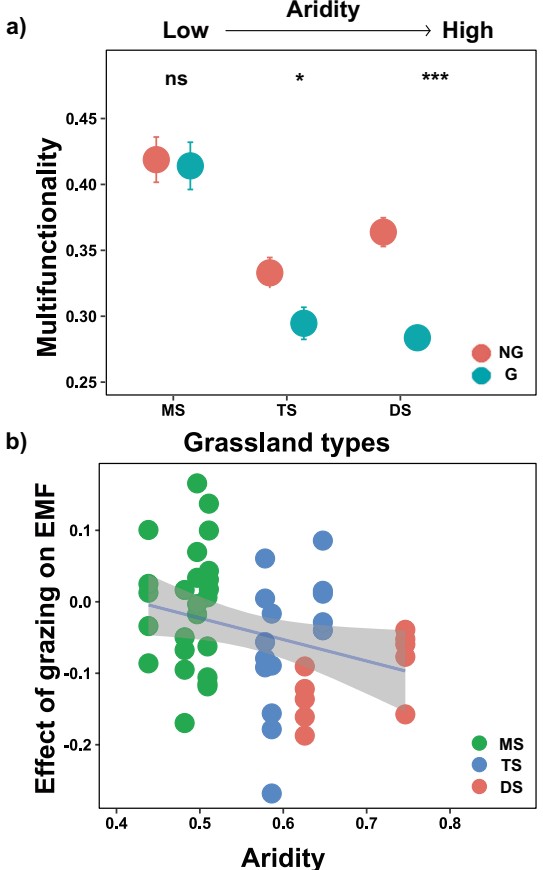

**Fig. 1 | The long-term effects of livestock grazing on multifunctionality across aridity gradient including three types of grasslands. a** Difference in multi-functionality inside (ungrazed) and outside (grazed) exclosure in three types of grasslands. Dots with bars indicate means ± standard error (SE) (MS: $n = 25$; TS: $n = 15$; DS: $n = 10$). Statistical analysis was performed using linear mixed effects models with grazing, grassland types and their interaction as fixed factors, and plot nested within sites as random factors; The two-tailed statistical tests indicate significant effects by $*P < 0.05$; $***P < 0.001$; ns, nonsignificant. For exact statistical values, see Supplementary Table 3. NG, no grazing; G, grazing. **b** Relationships between aridity and the effects of grazing on multifunctionality (two-sided Pearson adjusted r-squared = 0.069, $p = 0.036$). The solid line represents the linear regression, while the gray shading indicates the 95% confidence interval. Log response ratios (LRRs) compare multifunctionaliy outside and inside exclosures were used to quantify the effects of grazing on multifunctionality. EMF ecosystem multifunctionality, MS meadow steppe, TS typical steppe, DS desert steppe. Source data are provided as a Source Data file.

weighed multifunctionality (Supplementary Fig. 7), suggesting that the results are robust to methodological approaches. Also, our results were similar after accounting for environmental factors such as mean annual temperature (MAT) and mean annual precipitation (MAP) using Structural Equation Modeling (SEM) (Supplementary Fig. 10). Together, our results suggest that, in highly grazed ecosystems by domestic herbivore, plant cannot longer support more functions. In these ecosystems, soil biodiversity becomes the major driver of multifunctionality, highlighting the relevance of its conservation.

The shift in the contribution of biodiversity on multifunctionality could be attributed to the long-term grazing-induced change in plant composition (Supplementary Fig. 3a, b) and soil properties. Our results show that grazing increased the relative abundance of annual species (Supplementary Fig. 3a, b). These species are usually the acquisitive species, which has been proven to have a limited potential to increase ecosystem functioning such as biomass growth and storage, especially in dry conditions[39, 40]. Such change may result in the relative minor

effects of plant diversity on multifunctionality. Moreover, long-term continuous grazing-induced harsh environment may further contribute to these changes. Previous studies have shown that plant diversity has stronger effects on functioning in soils with higher nutrient[41], while the predominant role of soil biodiversity in regulating ecosystem functions has been proven important in harsh environments[28, 29, 42, 43]. Further analyses showed that soil fungal and protist diversity were especially important for supporting multifunctionality (Supplementary Fig. 9). Most soil protists are phagotrophic[44] and prey on bacteria and fungi, which leads to changes in microbial biomass, activity, and community structure[45]. Thus, the role of protists may have top-down trophic impacts when controlling ecosystem multifunctionality and regulating the importance of different trophic levels for supporting function[46, 47].

Our cross-site work provides solid empirical evidence that relatively long-term impacts of managed grazing depend on site aridity, with the most arid sites experiencing more negative impacts on ecosystem multifunctionality. Our findings also provide field experimental evidence that aridity exacerbates the negative effects of long-term grazing on belowground biodiversity and further strengthened their contributions to the decline in multifunctionality at arid grasslands. Thus, domestic herbivore grazing shifted the contribution of biodiversity drivers of multifunctionality from plant to soil biodiversity when moving from ungrazed to grazed ecosystems. We also suggest that future work further examine these changes over an even longer timescale and across an even larger spatial scale. The messages for the land managers and policymakers are clear: livestock grazing should be given more caution in more arid grasslands wherein multifunctionality is vulnerable to livestock grazing and highly dependent on soil biodiversity. Thus, conserving soil biodiversity is critical for supporting the multifunctionality in widespread grazed grassland worldwide. Our study also includes a series of implications for the management and conservation of grasslands under global change. Our results suggest that increased global warming and reduced precipitation may increase the risk of the negative effects of grazing on grassland biodiversity and function. To protect the long-term health of grasslands and avoid the degradation of grasslands in a changing world, we call for adjusting livestock numbers and the spatial geographical range suitable for grazing to adapt to a drier and warmer planet.

## Methods
### Experimental sites
This study was conducted in dry grasslands from Northern China (111.23 E to 123.51 E, 41.25 N to 49.52 N) ranging from arid to semi-arid drylands. We estimated the mean annual temperature, mean annual precipitation, and aridity level of each site using datafrom the World-Clim global database[48] (https://www.worldclim.org/). Mean annual precipitation varied from 232 mm to 435 mm, mean annual air temperature ranged from −2 °C to 5.4 °C, and the aridity ranged from 0.438 to 0.746. This region has experienced more than half a century of continuous grazing. The main grazing livestock are sheep and goats, followed by cattle and horses[49]. Traditionally, these grasslands have been used for grazing by nomadic tribes in a sustainable way. However, in the past 60–70 years, land use shifted from extensive grazing by nomadic pastoralists to intensive livestock production. These rangelands are used by ranchers with permits to public grazing allotments. Since the 1980s, these grasslands from Northern China were highly grazed resulting in important overgrazing and grassland degradation[49–52]. Stocking rates have dramatically increased from 0.3SE/ha in 1947 to 2.5 SE/ha in 2015 in this region[49]. We choose 10 experimental sites including three different types of grasslands situated along a 1100 km transect from east to west including meadow steppes, typical steppes, and desert steppes across an aridity gradient (from less to more arid; Table S1; Supplementary Fig. 1, 2). These sites represent three major grassland types including the most dominant

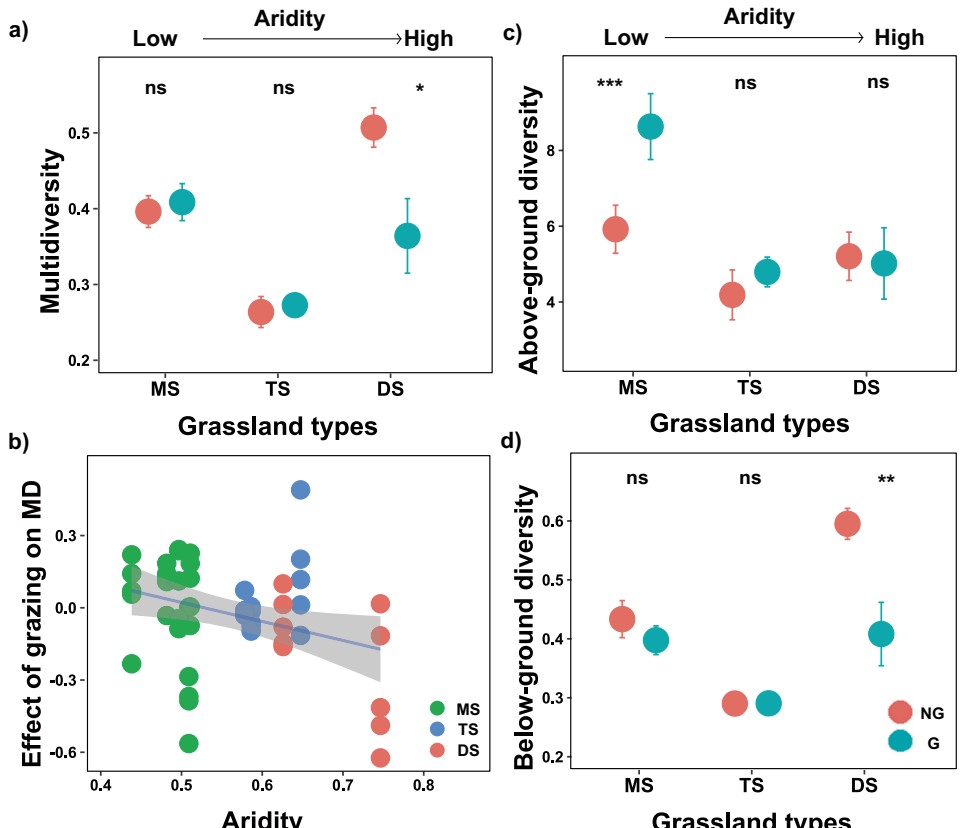

**Fig. 2 | The long-term effects of livestock grazing on biodiversity across aridity gradient including three types of grasslands.** Difference in **a** multidiversity, **c** above-ground diversity, and **d** below-ground diversity inside (ungrazed) and outside (grazed) exclosure in three types of grasslands. Dots with bars indicate means ± standard error (SE) (MS: $n = 25$; TS: $n = 15$; DS: $n = 10$). Statistical analysis was performed using linear mixed effects models with grazing, grassland types and their interaction as fixed factors, and plot nested within sites as random factors; The two-tailed statistical tests indicate significant effects by *$P < 0.05$; **$P < 0.01$;

***$P < 0.001$; ns, nonsignificant. For exact statistical values, see Supplementary Table 3. NG, no grazing; G, grazing. **b** Relationships between aridity and the effects of grazing on multidiversity (two-sided Pearson adjusted r-squared = 0.086, $p = 0.022$). The solid line represents the linear regression, while the gray shading indicates the 95% confidence interval. Log response ratios (LRRs) compare multi-diversity outside and inside exclosures were used to quantify the effects of grazing on multidiversity. MD multidiversity, MS meadow steppe, TS typical steppe, DS desert steppe. Source data are provided as a Source Data file.

vegetation characteristics found in northern China (Supplementary Table 1; Supplementary Fig. 1). All these sites have been in the over-grazing intensity.

We evaluated the relatively long-term effects of domestic herbivore grazing by excluding livestock for an extended period of time to get comparable control, which has been shown to be an effective treatment to examine the grazing effects[53–55]. At each experimental site, a > 10 years long-term grazing exclosure was established (Supplementary Table 1), and all these grasslands inside enclosure have greatly been different from that outside enclosure due to long-term prohibition of livestock grazing (Supplementary Fig. 1).

### Field sampling
At each site, a pair of sampling area (50 m × 50 m) was selected randomly on both sides of the fence, and 5 1 m × 1 m plots (5 replicates for control including grazing and 5 replicates for grazing exclusion) were set at the four corners and the center of the area. These replicates are included in our statistical models as random factors to avoid pseudo-replication (see statistical modeling below). Plant and soil sampling were carried out during the summer (late July to August) of 2020, corresponding to annual peak-standing biomass. Above-ground biomass was clipped at the ground level and oven dried at 65 °C for 48 h. Then it was weighed and ground into a fine powder on a ball mill for plant community nitrogen and phosphorus analyses. Soil samples were collected by taking five soil cores (2.5-cm diameter) at 10 cm depth in each of the five 1 × 1 m plots at each site. The five soil cores

were mixed in situ to form one composite sample representing each plot. After removing the rocks and roots, the soil was passed through a 2-mm-mesh sieve and separated into two parts. One part was air-dried and used to determine soil organic C. The other part was kept in a freezer (MOBICOOL CoolFreeze CF-50) to maintain a temperature of −18 °C and carried back to the laboratory as soon as possible for soil microbial community analysis and microbial biomass C, N, and available nitrogen analysis. We then collected belowground root biomass to a depth of 30 cm using soil cores (diameter 7 cm) in each of these five quadrats as well. Roots were collected by rinsing the samples using sieves (mesh size 0.25 mm) on the same day, and then oven-dried at 65 °C for 48 h and weighed.

### Soil biodiversity
**Sequencing.** We used next generation sequencing technology to characterize the biodiversity of soil bacteria, fungi and protists. Microbial DNA was extracted from soil samples using the PowerSoil DNA Isolation Kit (Mo Bio Laboratories, Carlsbad, CA, USA) according to the manufacturer's protocols. The final DNA concentration and purification were determined by the NanoDrop 2000 UV-vis spectro-photometer (Thermo Scientific, Wilmington, USA). Bacterial communities were assessed with primers 338 F (5′-ACTCCTACGGGAG GCAGCAG-3′) and 806 R (5′-GGACTACHVGGGTWTCTAAT-3′), targeting the V3–V4 regions of the 16 S rRNA gene. Fungal communities were assessed using the forward primer ITS-1F (5′-CTTGGTCATTTA-GAGGAAGTAA-3′) and the reverse primer ITS-2R (5′-GCTGCGTTC

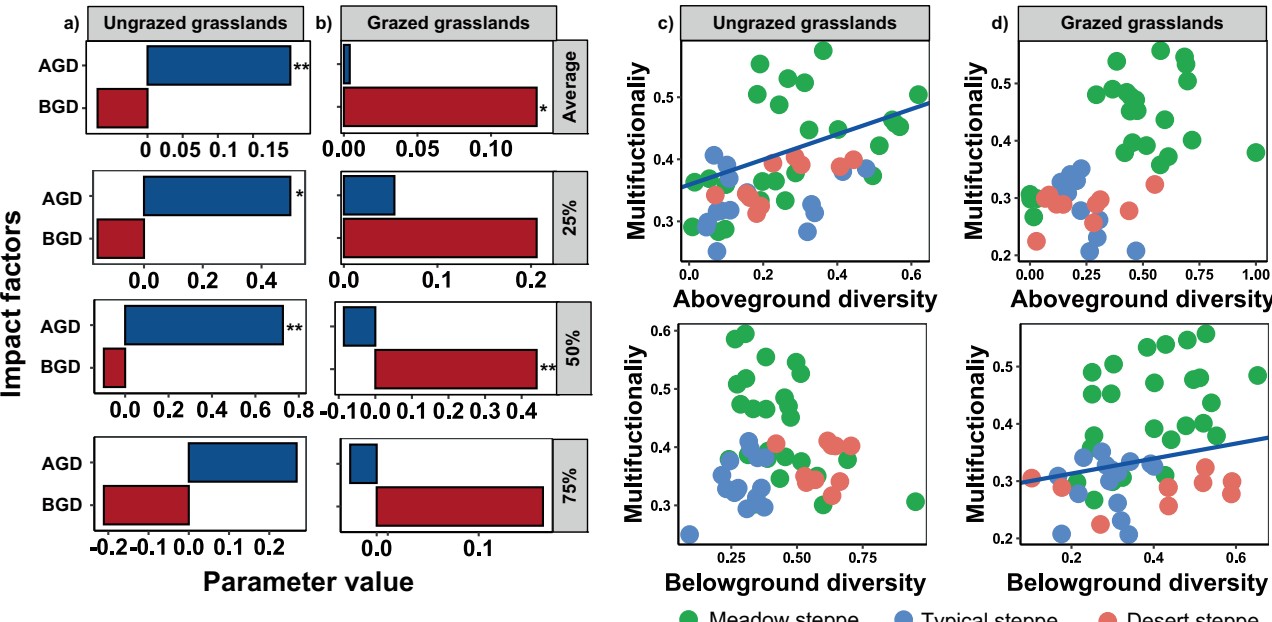

**Fig. 3 | The relative strength of above- and below-ground diversity in supporting multifunctionality in ungrazed grasslands and long-term grazed grasslands.** The relative strength of above- and below-ground diversity for average multifunctionality and multithreshold functioning (the number of functions above multiple thresholds) in ungrazed grasslands **a** and grazed grasslands **b**. And the fitted linear relationships between average multifunctionality and above-ground diversity and below-ground diversity in ungrazed grasslands **c** and grazed grasslands **d**. All statistical analysis was performed using linear mixed-effects models with aboveground and belowground diversity as fixed factors, and plots and sites nested within grassland types as random factors ($n = 50$); The two-tailed statistical tests indicate significant effects by *$P < 0.05$; **$P < 0.01$. For exact statistical values, see Supplementary Table 5. The aboveground diversity and the numbers of functions beyond a given threshold (25%, 50%, and 75%) were standardized (min-max normalization) variables before the analysis. AGD above-ground diversity, BGD below-ground diversity, Average the average EMF, 25%, 50%, and 75%, the number of functions beyond 25%, 50%, and 75% threshold. Source data are provided as a Source Data file.

TTCATCGATGC-3′). To characterize the diversity of protists, the hypervariable V4 region of the 18 S rRNA gene was amplified using the general eukaryotic primers TAReuk454F (5′-CCAGCASCYGCGG-TAATTCC-3′) and TAReukREV3R(5′-ACTTTCGTTCTTGATYRA-3′). All PCR reactions were conducted using the following program: 3 min of denaturation at 95 °C, 27 cycles (bacterial) or 35 cycles (fungal) and 30 s at 95 °C, 30 s at 55 °C for annealing, 45 s at 72 °C for elongation, and a final extension at 72 °C for 10 min. PCR reactions were performed in triplicate 20 μL mixture containing 4 μL of 5 × FastPfu Buffer, 2 μL of 2.5 mM dNTPs, 0.8 μL of each primer (5 μM), 0.4 μL of FastPfu Polymerase and 10 ng of template DNA. The resulting PCR products were extracted from a 2% agarose gel, further purified using the AxyPrep DNA Gel Extraction Kit (Axygen Biosciences, Union City, CA, USA), and quantified using QuantiFluor™-ST (Promega, USA) according to the manufacturer's protocol. Purified amplicons were pooled in equimolar and paired-end sequences on an Illumina MiSeq platform (Illumina, San Diego, USA).

**Bioinformatics.** The MiSeq sequences were demultiplexed and quality-filtered by Trimmomatic on the criteria of having an average quality score higher than 20 over a 50 bp sliding window. Sequences whose overlap was longer than 10 bp were merged according to their overlap sequence. After removing the reads containing ambiguous bases, paired-end reads with at least a 10 bp overlap were joined using FLASH and allowing for 2 mismatched nucleotides. Operational taxonomic units (OTUs) were clustered with a 97% similarity cutoff using UPARSE[56]. Singleton OTUs were removed as well as the chimeric sequences identified by the UCHIME algorithm. For 16 S, taxonomy was assigned using the SILVA reference database 138[57]. For ITS, the taxonomy assignment was performed using the UNITE reference database[58] (v.8.0). For 18 S, taxonomy was assigned using PR2 database 4.5[59]. Based on taxonomic assignments, we filtered out OTUs from

the 18 S rRNA gene sequences that were non-protist (i.e., OTUs belonging to Fungi, Streptophyta, and Metacoa). The relative abundance of saprotrophs, mutualistic fungi, and potential fungal plant pathogens in soils were obtained from the amplicon sequencing analyses (as explained above) and was inferred by parsing the soil phylotypes using FungalTraits[60]. The inverse abundance (reduced relative abundance) of potential fungal plant pathogens was obtained by calculating the inverse of this variable (total relative abundance of fungal plant pathogens×−1).

**Biodiversity indices.** We calculated both species richness and a diversity index that included both richness and evenness for plants, soil bacteria, fungi, and protists. Species richness was calculated as the total number of species in the quadrat. Diversity was calculated using the exponential of Shannon's entropy[61], exp (H′):

$$H' = -\sum_{i=1}^{S} (P_i \ln P_i)$$

where Pi is the proportional abundance of species I, summed for all the species measured. Moreover, we combined the soil biodiversity characteristics (soil bacterial, fungal, and protist diversity) by averaging the standardized scores (min-max normalization) of diversity to obtain a single index reflecting the below-ground diversity and below-ground species richness. The same method was used to obtain the ecosystem biodiversity (Multidiversity, multirichness) by combining the above-ground and below-ground biodiversity.

### Ecosystem functions

We used 11 variables reflecting ecosystem multifunctionality including above-ground biomass, below-ground biomass, plant community N, plant community P, soil organic C, soil available N, microbial biomass

C, microbial biomass N, decomposers, pathogen control, and mycorrhizal mutualism (see Supplementary Table 2, Supplementary Figs. 4, 5 for further rationale on the selected functions). Soil organic C was determined with the K2Cr2O7 titration method after digestion. Soil microbial biomass carbon and microbial biomass nitrogen were measured by the chloroform fumigation-extraction method. Soil NH$_4^+$ and NO$_3^-$ were analyzed using an Alliance Flow Analyzer (Alliance Flow Analyzer, Futura, frépillon, France). Soil available N was determined as the sum of ammonium and nitrate. Plant community N content was measured using the CHNOS Elemental Analyzer (vario EL cube), and phosphorus content was analyzed using fully automated high-technology discrete analyzer (Smartchem 450, AMS, France) after H$_2$SO$_4$·H$_2$O$_2$ digestion.

### Assessing ecosystem multifunctionality

We used multiple methods to determine ecosystem multifunctionality, from averaging and weighted multifunctionality to multi-threshold multifunctionality. By doing so, we aimed to cover different aspects of multifunctionality. First, we calculated the average multifunctionality (EMF), which is widely used in the multifunctionality literature[29]. We then calculated the weighted EMF to down-weight highly correlated functions ($r > 0.7$, $P < 0.001$; Supplementary Fig. 4) as described in Manning et al. 2018[30]. Importantly, the weighted EMF was highly correlated with the average multifunctionality ($r^2 = 0.937$, $P < 0.001$; Supplementary Fig. 6). These analyses suggest that the choice of multifunctionality index do not alter our results. Moreover, we further calculated the number of functions beyond a given threshold (25%, 50%, and 75%) using the multithreshold approach described in Byrnes et al. 2014[31], as explained in DelgadoBaquerizo et al. 2020[26]. Before analyses, all individual ecosystem function (EF) variables were standardized by transformation as follows: EF = (rawEF-min(rawEF))/(max(rawEF)−min(rawEF)), with EF indicating the final (transformed) ecosystem function value and raw EF indicating raw (untransformed) ecosystem function values. This way each transformed EF variable had a minimum value of zero and a maximum of 1.

### Statistical analysis

We conducted linear mixed effects models (LMMs) to analyze the interactive effects of grazing and grassland types (i.e., closely related to aridity) on ecosystem biodiversity, aboveground biodiversity (plant diversity), belowground biodiversity (bacterial diversity, fungal diversity, and protist diversity), and ecosystem multifunctionality. Grazing, grassland types, and their interaction were fitted as fixed factors, and plots (original experimental replicates) nested within the site were fitted as random factors to control for pseudo-replication. Further, we also fitted regression models to explore the relationship between aridity and the effects of grazing on biodiversity and ecosystem multifunctionality across 10 sites, respectively. The effects of grazing on biodiversity and ecosystem multifunctionality were estimated as the change in these factors inside and outside the exclosures. Change in each factor was estimated as the log ratio of the treatment divided by the control, log (Sf1/Sf2), where Sf1 is the biodiversity or multifunctionality in grazed plots and Sf2 is the biodiversity or multifunctionality in ungrazed plots. Considering specific years of enclosure was not exactly same for the ten sites despite they all have over 10-year enclosure history, we also used linear mixed effects models to analyze the effects of exclosure year on the grazing effects on EMF, multidiversity, above-ground diversity, and below-ground diversity. Exclosure year was taken as fixed factor. Plots and sites nested within grassland types were taken as random factor. The results showed that the difference in years of enclosure does not affect the responses of multifunctionality, multidiversity, above-ground diversity, and below-ground diversity to grazing (Supplementary Table 4).

We further determined the relative importance of plant diversity and soil microbial diversity (bacterial diversity, fungal diversity, and protist diversity) for ecosystem multifunctionality in non-grazed grasslands and grazed grasslands, respectively using linear mixed effects models. In these models, plant diversity and soil microbial diversity were fitted as fixed factors, and plot and site nested within grassland types were fitted as random factors. This analysis also conducted for the single functions and the possible combinations among functions. The plant diversity, soil bacterial diversity, soil fungal diversity, soil protists diversity, and the numbers of functions beyond a given threshold (25%, 50%, and 75%) were standardized (min-max normalization) variables before the analysis. Linear mixed effects modeling was also used to examine the grazing effects on single functions including above-ground biomass, below-ground biomass, plant community N, plant community P, soil organic C, soil available N, microbial biomass C, microbial biomass N, decomposers, pathogen control, and mycorrhizal mutualism with grazing as fixed factors, and plot and site nested within grassland type as random factors. Linear mixed effects modeling was performed using the *lme* function within the *nlme* package. We further controlled our results by any influence of climate, mean annual temperature (MAT) and mean annual precipitation (MAP). To such an end, we used *piecewise SEM*[62] on linear mixed-effects models including site and plot nested within grassland types as random factors. Our structural equation modeling was carried out using the *psem* function of the *piecewiseSEM* package. Non-metric multidimension scaling (NMDS) approach with Bray-Curtis distancematrix and PERMANOVA (Adonis) jointly were used to illustrate the differences in plant and microbial community structure between non-grazed and grazed grasslands. Post hoc pairwise adonis test were done using the function *pairwise. adonis* from package *pairwiseAdonis* with Bonferroni correction. All analysis were performed using R software (version 4.1.0).

### Reporting summary

Further information on research design is available in the Nature Portfolio Reporting Summary linked to this article.

## Data availability

All Bacterial, fungal and protist sequences have been deposited in NCBI's SRA database under project accession number PRJNA995873. All data that support the findings of this study are available in the Figshare database (https://doi.org/10.6084/m9.figshare.23713719)[63]. The mean annual temperature, mean annual precipitation, and aridity level of each site using data from the WorldClim global database[48] (https://www.worldclim.org/). Source data are provided in this paper.

## Code availability

R scripts used for data analysis have been deposited in the Figshare database (https://doi.org/10.6084/m9.figshare.23713719)[63].

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

## Acknowledgements

Ling Wang acknowledges support from National Key Research and Development Program of China (2022YFF1300604), the National Natural Science Foundation of China (No. 32271642), and the Program for Introducing Talents to Universities (B16011). Ministry of Education Innovation Team Development Plan, Grant/Award Number: 2013-373. Manuel Delgado-Baquerizo acknowledges support from the Spanish Ministry of Science and Innovation (PID2020-115813RA-I00), and a project of the Fondo Europeo de Desarrollo Regional (FEDER) and the Consejería de Transformación Económica, Industria, Conocimiento y Universidades of the Junta de Andalucía (FEDER Andalucía 2014–2020 Objetivo temático "01 - Refuerzo de la investigación, el desarrollo tecnológico y la innovación") associated with the research project P20_00879 (ANDABIOMA).

## Author contributions

M.Z. performed the experiments and led in data analysis and writing efforts; L.W. designed the experiments and contributed to data analysis and writing; M.D.-B., F.I., and Y.H. contributed to revise the manuscript critically; G.L., J.C., Y.W., Y.W., X.P., and Y.X. performed the experiments.

## Competing interests

The authors declare no competing interests.
