## [Peer Review File · Nature Communications]

Reviewers' Comments:

Reviewer #1:

Remarks to the Author:

In this article, the authors report on findings from grazing exclusion experiments. The experiments established and maintained pairs of grazed and ungrazed plots for ten or more years that were positioned in three major grassland types in northern China. The main findings emphasized that impacts from grazing estimated in this fashion depended on aridity level, "with the most arid sites experiencing more negative impacts on ecosystem multifunctionality."

I find this to be a solid empirical study that adds to our knowledge of the subject. There are a number of details from the study and suggest interesting mechanistic hypotheses, which is an added bonus to this work.

Generally, I am pleased with the work. There are a number of editorial changes needed, but most are very minor. The one thing that bothered me throughout was the description of the work as estimating "the long-term impacts of grazing". As described in the Methods, the grasslands have all been subject to long-term grazing impacts, with livestock grazing apparently intensifying 60-70 years prior to the study. Livestock exclusions were initiated some time prior to 2010, as best as I can tell. Given this, I would argue the title should be changed to something like,

"Impacts of release from long-term grazing on grassland biodiversity and function"

or

"Impacts from prolonged protection from grazing ..." if they wanted to continue to emphasize the duration of the experiment.

As things stand, the authors are arguing that 10 years is long-term and place great emphasis on that period of time. Still, that 10-year period is not long-term grazing, so their phrasing is not quite correct. From my perspective, we will never know the actual impacts of long-term grazing because that would require comparable ungrazed grasslands, which are not studied here. Perhaps this is a quibble, but it bothered me throughout my reading of the manuscript.

Editorial Suggestions:

1. Line 26 - Replace "herbivore" with "herbivores".
2. Line 31 - Drop "in" and replace "of" with "from".
3. Line 32 - Replace "of" with "on".
4. Line 36 - Replace "shifted" with "changed".
5. Line 93 - Replace "on" with "of".
6. Line 95 - Replace "the time-lag" with "time-lags in the".
7. Line 115 - Replace "done" with "maintained".
8. Line 119 - "We aimed to examine the long-term effects of ... grazing". Add, "by excluding grazers for an extended period of time."
9. Line 147 - Replace "function" with "functions".
10. Lines 150-152 - "This knowledge is essential to determine cause-effect relationships, which cannot be established from observational data". Please remove this sentence. While it is true in many cases, as an absolute statement, this is incorrect. There are a variety of ways that causal relationships can be established. How do we know that long-term smoking increases the risk of lung cancer? Not from experimental studies, but from the accumulation of mechanistic knowledge. Same with all of astronomy and astrophysics. The subject of building causal knowledge is complicated and it is unhelpful for folks to make simplistic declarations - one of the reasons ecologists are afraid of the subject and know so little about it.

Aside from these details, I find this to be a useful contribution to the literature.

Reviewer #2:

Remarks to the Author:

Grassland ecosystems provide a huge amount of ecosystem services for human beings. However,

accelerated rates of global change (such as changes in precipitation) and anthropogenic activities (such as overgrazing) are altering the structure and function of grassland ecosystems worldwide. I am very interested in the paper titled "Long-term impacts of grazing on grassland biodiversity and function are driven by aridity" by Minna Zhang et al. submitted to Nature Communications for consideration of publication. The strengths of the paper are as follows: 1) This study takes advantage of paired grazed and ungrazed plots across a large gradient of aridity and grassland types, including desert steppe, typical steppe and meadow steppe. Doing so, the combined effects of aridity (an index integrating precipitation and temperature) and livestock grazing on the above- and below-ground biodiversity could be simultaneously explored; 2) The paired plots for taking samples of plants, soils and soil organisms were geographically very close and the initial floristic composition was similar. However, they had experienced different grazing intensity for over 10 years. This provided an opportunity to study the long-term impacts of livestock grazing on the structure and function in the largest, contiguous Eurasian steppe ecosystems encompassing three major grassland types for livestock grazing; 3) The authors explored the combined impacts of aridity (Dryness) and animal grazing on ecosystem multifunctionality, for which Manuel Delgado-Baquerizo, a co-author of the paper, is an expert in the field; 4) This is a novel approach that involves measurement of an important ecosystem function metric by considering so many biotic and abiotic variables, from genomic to organismal levels. Particularly, I am impressed by their statistical analysis and computation of different multifunctionality indices.

However, I have several concerns on the paper. First, the paired plots had experienced different grazing intensities over 10 years, but they only took field measurements in a single year at the time of peak biomass. In the Inner Mongolia Plateau, annual variations of biomass production can be very large due to fluctuating environmental factors such as precipitation and temperature, as well as insect herbivory and rodent damage. Second, they did not provide any information on the types of grazing animals and the stocking rates, all of which had different impact on the multifunctionality. For the above two questions, they can get the information from published data in literature for this transect to indicate that one-time sampling can reflect the means of the major variables they measured. Third, they took soil samples from the top 10cm only, without considering the subsoil. I believe that in the continuously grazed plots, some of the initial topsoil might have had eroded away by wind, particularly in the desert steppe, therefore subsoil samples are recommended to be reported, at least point out this problem in the revised version of the paper. Fourth, they only used to Shannon's Index to represent biodiversity. I suggest they also need to report the species richness (number of species). For the biodiversity metric, two indices are needed: One representing richness and another representing evenness or dominance.

Also, the English grammar needs to be checked carefully. For example,

Line 31 Change "in paired of grazed and ungrazed" to "in paired grazed and ungrazed"

Line 37 Change "above and belowground" to "above- and below-ground"

Line 108 Change "capacity" to "ability"

Line 139 Change "nutrient plant uptake" to "plant nutrient uptake"

Line 170 Change "Our study found" to "We found"

Reviewer #3:

Remarks to the Author:

This study examines the long-term effects of ungulates grazing on biodiversity and ecosystem multifunctionality (EMF) through a network of 10 experimental sites on a gradient of aridity.

Although there are some large-scale studies on the impact of grazing on biodiversity and ecosystem multifunctionality, there are few studies based on consistent sampling methods. As this is a multi-site study, this is where my concerns lie.

First, this is a multi-site comparative study, not strictly an experiment of the same standard. As can be seen from the study methodology, a comparison between in-fenced and free grazing was chosen for each site, so it was a pseudo-replicated experimental design at each site. As the authors state, there is an urgent need for a multi-site, regional standardized experiment to investigate the effects of grazing on biodiversity and multiple ecosystem functions to experimentally determine the causal relationships between grazing, biodiversity and multifunctionality. Unfortunately, this study cannot serve that purpose.

Second, for grazing experiments, the most important factors to consider are the grazing history of the study site, the animals grazed, the intensity of grazing, the season and the duration of grazing. Unfortunately, these details were not available for each site. These factors have a direct impact on the understanding and interpretation of the results of the study. For example, the manuscript simply states that all study sites are enclosed for >10 years, but does not provide a specific enclosure time.

Third, in the statistical analysis, the relationship between biodiversity and multifunctionality was studied without controlling for the influence of environmental variables, which may have overestimated the effect of plant and soil microbial diversity on multifunctionality.

Fourth, if this is a long-term experiment, the analysis should not be limited to the final results. Background values at the beginning of the experiment, data during the experiment, even if there are multiple sampling at some sites, can be very helpful in the analysis of the results.

Some minor issues:

(1) Site 9, with Aridity index of 0.626, within the typical grassland range in the MS. Why is that?

(2) Table S1 should have detailed information for each site.

Reviewer #1 (Remarks to the Author):

In this article, the authors report on findings from grazing exclusion experiments. The experiments established and maintained pairs of grazed and ungrazed plots for ten or more years that were positioned in three major grassland types in northern China. The main findings emphasized that impacts from grazing estimated in this fashion depended on aridity level, “with the most arid sites experiencing more negative impacts on ecosystem multifunctionality.”

I find this to be a solid empirical study that adds to our knowledge of the subject. There are a number of details from the study and suggest interesting mechanistic hypotheses, which is an added bonus to this work.

1. We appreciate the positive and constructive comments by the reviewer. We are glad to read that the reviewer believes that our manuscript would be a solid empirical study that adds to our knowledge of the subject. Responses to all comments are below – we found these comments very useful as they helped us improve the quality of the manuscript.

Generally, I am pleased with the work. There are a number of editorial changes needed, but most are very minor. The one thing that bothered me throughout was the description of the work as estimating “the long-term impacts of grazing”. As described in the Methods, the grasslands have all been subject to long-term grazing impacts, with livestock grazing apparently intensifying 60-70 years prior to the study. Livestock exclusions were initiated some time prior to 2010, as best as I can tell. Given this, I would argue the title should be changed to something like,

“Impacts of release from long-term grazing on grassland biodiversity and function”

or

“Impacts from prolonged protection from grazing ...” if they wanted to continue to emphasize the duration of the experiment.

As things stand, the authors are arguing that 10 years is long-term and place great emphasis on that period of time. Still, that 10-year period is not long-term grazing, so their phrasing is not quite correct. From my perspective, we will never know the actual impacts of long-term grazing because that would require comparable ungrazed grasslands, which are not studied here. Perhaps this is a quibble, but it bothered me throughout my reading of the manuscript.

2. Thanks. We fully understand the concern of the reviewer. We agree that retrieving comparable ungrazed grasslands is not an easy task. We did our best to obtain ungrazed grasslands, here over a decade of grazing exclusion grasslands. In fact,

it is quite difficult and valuable finding these types of controlled exclusions across contrasting climatic conditions, and this is one of the strongest points in our manuscript. We now updated our title to remove the word “long-term” following the reviewer’s suggestion. We also emphasized the experimental nature of grazing impact compared with previous observational studies. We also toned down our claims on long-term research as suggested and added further discussions about this important point (lines 237-238).

Editorial Suggestions:

1. Line 26 - Replace “herbivore” with “herbivores” .
2. Line 31 - Drop “in” and replace “of” with “from” .
3. Line 32 - Replace “of” with “on” .
4. Line 36 - Replace “shifted” with “changed” .
5. Line 93 - Replace “on” with “of” .
6. Line 95 - Replace “the time-lag” with “time-lags in the” .
7. Line 115 - Replace “done” with “maintained” .
8. Line 119 - “We aimed to examine the long-term effects of … grazing”. Add, “by excluding grazers for an extended period of time.”
9. Line 147 - Replace “function” with “functions” .

3. Thank you for these edits. All these have been revised.

Lines 150-152 - “This knowledge is essential to determine cause-effect relationships, which cannot be established from observational data” . Please remove this sentence. While it is true in many cases, as an absolute statement, this is incorrect. There are a variety of ways that causal relationships can be established. How do we know that long-term smoking increases the risk of lung cancer? Not from experimental studies, but from the accumulation of mechanistic knowledge. Same with all of astronomy and astrophysics. The subject of building causal knowledge is complicated and it is unhelpful for folks to make simplistic declarations - one of the reasons ecologists are afraid of the subject and know so little about it.

4. Thanks. This is valid point and we have removed this sentence as the reviewer suggested.

Aside from these details, I find this to be a useful contribution to the literature.

5. We really appreciated for your positive comments and some good suggestion for our study.

Reviewer #2 (Remarks to the Author):

Grassland ecosystems provide a huge amount of ecosystem services for human

beings. However, accelerated rates of global change (such as changes in precipitation) and anthropogenic activities (such as overgrazing) are altering the structure and function of grassland ecosystems worldwide. I am very interested in the paper titled “Long-term impacts of grazing on grassland biodiversity and function are driven by aridity” by Minna Zhang et al. submitted to Nature Communications for consideration of publication. The strengths of the paper are as follows: 1) This study takes advantage of paired grazed and ungrazed plots across a large gradient of aridity and grassland types, including desert steppe, typical steppe and meadow steppe. Doing so, the combined effects of aridity (an index integrating precipitation and temperature) and livestock grazing on the above- and below-ground biodiversity could be simultaneously explored; 2) The paired plots for taking samples of plants, soils and soil organisms were geographically very close and the initial floristic composition was similar. However, they had experienced different grazing intensity for over 10 years. This provided an opportunity to study the long-term impacts of livestock grazing on the structure and function in the largest, contiguous Eurasian steppe ecosystems encompassing three major grassland types for livestock grazing; 3) The authors explored the combined impacts of aridity (Dryness) and animal grazing on ecosystem multifunctionality, for which Manuel Delgado-Baquerizo, a co-author of the paper, is an expert in the field; 4) This is a novel approach that involves measurement of an important ecosystem function metric by considering so many biotic and abiotic variables, from genomic to organismal levels. Particularly, I am impressed by their statistical analysis and computation of different multifunctionality indices.

6. We appreciate the positive and constructive comments by the reviewer, who have nicely captured some of our key points. Responses to all comments are below – we found these comments very useful as they helped us improve the quality of the manuscript and better highlight our key findings.

However, I have several concerns on the paper. First, the paired plots had experienced different grazing intensities over 10 years, but they only took field measurements in a single year at the time of peak biomass. In the Inner Mongolia Plateau, annual variations of biomass production can be very large due to fluctuating environmental factors such as precipitation and temperature, as well as insect herbivory and rodent damage.

7. Thanks. We fully understand the reviewer’s concern. We agree that factors such as plant biomass can fluctuate among years, however, given the large spatial scale of our survey we expected a reduced impact on our results. Simply put, more arid locations will always have less biomass than less arid locations regardless of local temporal variation. There is an inevitable tradeoff between having long-term experimental studies at few sites or short-term experimental studies at many sites. Our study has the strengths and limitations of the latter, similar to many other multi-site

studies (e.g., Fanin et al. 2018 Nat Ecol Evol; Chen et al. 2018 PNAS; Hu et al. 2021 Nature communications). To provide further evidence that our results are robust to this important point, we added another year of biomass data, recalculated our EMF index using the average above-ground biomass of 2018 and 2020, and found similar results (see lines 193-199; Fig. S11, S12). We thank the reviewer for suggesting that we strengthen our paper in this way.

Second, they did not provide any information on the types of grazing animals and the stocking rates, all of which had different impact on the multifunctionality. For the above two questions, they can get the information from published data in literature for this transect to indicate that one-time sampling can reflect the means of the major variables they measured.

8. We appreciate reviewer's understanding and good suggestion to resolve the two questions. The area outside the fences have experienced more than half a century of continuous complicated grazing history, and grazing intensity through time has not been constant. For example, previous publications have shown that stocking rates have dramatically increased from 0.3SE/ha in 1947 to 2.5 SE/ha in 2015 in this region (Kemp et al. 2020). The main grazing livestock are sheep and goats, followed by cattle and horses (Kemp et al. 2020). Based on the published data in the literature for this region, we can get the qualitative grazing intensity indicating that all these sites were overgrazed since the 1980s (Tong et al. 2004; Müller 2009; Wang et al. 2016; Bryan et al. 2018; Kemp et al. 2020). We agree that differences in grazing intensity among sites may explain part of the result. We would like to note that in order to reduce possible bias due to differences in grazing intensity, all sites selected in our study have been in the overgrazing intensity, which exceed ecosystem capacity threshold, thus small changes in grazing intensity may not change the state based on threshold models. We sincerely hope to get the reviewers and editor's understanding for these points. To take into account this important point we have now supplied more details about the grazing history in the methods section in our revised manuscript (see lines 259-274).

Third, they took soil samples from the top 10 cm only, without considering the subsoil. I believe that in the continuously grazed plots, some of the initial topsoil might have had eroded away by wind, particularly in the desert steppe, therefore subsoil samples are recommended to be reported, at least point out this problem in the revised version of the paper.

9. We appreciated the reviewer's point. We have added further discussions about this point in our revised manuscript (see lines 169-173).

Fourth, they only used to Shannon's Index to represent biodiversity. I suggest they also need to report the species richness (number of species). For the biodiversity metric, two indices are needed: One representing richness and another representing evenness or dominance.

10. We thank the reviewer for pointing out to this important aspect. We have added the analysis about species richness (lines 357-368; Table S3), and added further discussions about this important point (lines 161-164). Given the results for ecosystem richness was not significant, we keep using our original Shannon entropy biodiversity index in the main text, as this index can reflect both the number of species present and their relative abundances, and results about richness were added in supplementary files.

**Also, the English grammar needs to be checked carefully. For example,
Line 31 Change “in paired of grazed and ungrazed” to “in paired grazed and ungrazed”
Line 37 Change “above and belowground” to “above- and below-ground”
Line 108 Change “capacity” to “ability”
Line 139 Change “nutrient plant uptake” to “plant nutrient uptake”
Line 170 Change “Our study found” to “We found”**

11. Thank you for these edits. All these have been revised, and the English grammar has been checked carefully throughout the manuscript.

Reviewer #3 (Remarks to the Author):

This study examines the long-term effects of ungulates grazing on biodiversity and ecosystem multifunctionality (EMF) through a network of 10 experimental sites on a gradient of aridity. Although there are some large-scale studies on the impact of grazing on biodiversity and ecosystem multifunctionality, there are few studies based on consistent sampling methods. As this is a multi-site study, this is where my concerns lie.

12. We appreciate reviewer to point out all critical points and some constructive comments, which helped us solid our results and improve the quality of the manuscript. We agree that previous large-scale datasets were based on observational data or non-standardized surveys. We would like to take this opportunity to clarify that our surveys were conducted following relatively standardized protocols across the experimental sites, making our data more directly comparable than in previous studies. We now clarified this important aspect in lines 89-97, 115-121. Responses to all comments are below.

First, this is a multi-site comparative study, not strictly an experiment of the same standard. As can be seen from the study methodology, a comparison between in-fenced and free grazing was chosen for each site, so it was a pseudo-replicated experimental design at each site. As the authors state, there is an urgent need for a multi-site, regional standardized experiment to investigate the effects of grazing on biodiversity and multiple ecosystem functions to

experimentally determine the causal relationships between grazing, biodiversity and multifunctionality. Unfortunately, this study cannot serve that purpose.

13. We are grateful to the reviewer for pointing out this important confusion. We agree that our study is a multi-site comparative study, not strictly an experiment of the same standard. We are now using “multi-site comparable experiment” instead of “standardized experiment”, to highlight the experimental nature of our work compared with previous observational studies where grazed and ungrazed grassland distributed in different sites, or meta-analyses with unstandardized sampling and analytical approaches. Anyway, an ideal and more standardized experiment should be needed for this kind of regional scale multi-sites study. However, we really hope to get the reviewer’s understanding that this comparable regional scale study had greatly move more standardized experiment to help us explore effects of long-term livestock grazing on grasslands. As we know, it’s almost impossible to know the actual long-term impacts of grazing because that would require comparable never ungrazed grasslands, which is not possible in most regions of the planet. Here, we did our best to obtain ungrazed grasslands by excluding grazers for a relatively extended period of time. We used these controlled exclusions to identify an aridity gradient with and without exclusions and then respond our research questions. In fact, it is quite difficult and valuable finding these types of controlled exclusions across contrasting climatic conditions, and this is one of the strongest points in our manuscript.

About the pseudo-replication. Indeed, the five sampling replicates within each site are pseudo-replicated, but note that each grassland types have multiple experimental sites replications. This design was already considered in our statistical analyses to avoid statistical pseudo-replication and to capture the spatial heterogeneity within our plots (see lines 410-412; 432-434).

All these have been clarified and explained (see lines 89-97; 115-121; 410-412; 432-434).

Second, for grazing experiments, the most important factors to consider are the grazing history of the study site, the animals grazed, the intensity of grazing, the season and the duration of grazing. Unfortunately, these details were not available for each site. These factors have a direct impact on the understanding and interpretation of the results of the study. For example, the manuscript simply states that all study sites are enclosed for >10 years, but does not provide a specific enclosure time.

14. We fully understand the reviewer’s concern about the grazing history. Please, see response #9, where we address this important point, which was also made by reviewer #2. We also apologize for the unclear information, and now we supplied more details and relative explanation about the grazing history in methods (see lines 259-274). In addition, we have provided the specific enclosure year of each site in Table S1. We further examined whether the specific years of enclosure affected the grazing effects (lines 419-427), and the results showed that the specific years of

exclosure do not affect the responses of multifunctionality, multidiversity, above-ground diversity, and below-ground diversity to grazing (Table S4). More importantly, our experimental design supported the approach to address our research questions, and provided a unique opportunity to investigate the role of grazing enclosures in regulating biodiversity and function across an aridity gradient in one of the largest remaining grasslands of the planet. This knowledge is novel and important to ensure the sustainability of these fundamental ecosystems by providing empirical evidence on under what aridity conditions grazing can impact biodiversity and function.

Third, in the statistical analysis, the relationship between biodiversity and multifunctionality was studied without controlling for the influence of environmental variables, which may have overestimated the effect of plant and soil microbial diversity on multifunctionality.

15. This is a good point. We have evaluated for the importance of plant and soil microbial diversity as predictors of EMF while simultaneously accounting for the mean annual precipitation (MAP) and temperature (MAT) which has been provided to be the most important environmental variables mediating biodiversity effects (Jing et al. 2015, Nat. Commun; Hu et al. 2021, Nat. Commun) using structure equal models (SEM) as explained in lines 444-449. Our results were retained after controlling for these potential confounding variables (Fig. S10). We have now clarified this important point (see lines 199-202; 444-449).

Fourth, if this is a long-term experiment, the analysis should not be limited to the final results. Background values at the beginning of the experiment, data during the experiment, even if there are multiple sampling at some sites, can be very helpful in the analysis of the results.

16. We understand the reviewer's concern about final result from one-year data. However, considering that this is a regional scale multi-site comparative study of ecosystem multifunctionality (i.e., many measured response variables), and also considering this study aims to examine the long-term effects of livestock grazing (an important global change in grasslands worldwide) where we need exclude grazer for relatively extended period of time as much as possible to get comparable ungrazed grassland, we only show the final results from maximization time. As we know, large-scale multiple-site experiment commonly spanned enough climate gradients, the inter-annually climate variation may have relative minor effects on the results, we also noted that the one year data have been usually adopted by most multi-site studies (e.g., Fanin et al. 2018 Nat Ecol Evol; Chen et al. 2018 PNAS; Hu et al. 2021 Nature communications). To provide further evidence that our results are robust to this important point, and considering that biomass production may be the most sensitive factor that may change among years, we have recalculated our EMF index using the average above-ground biomass of 2018 and 2020. Importantly, similar results were

found (see lines 193-199; Fig. S11, S12). We believe that this point was very important and help us to conduct these new analyses which now contributed to the robustness of our work.

Some minor issues:

(1) Site 9, with Aridity index of 0.626, within the typical grassland range in the MS. Why is that?

17. We apologize for the confusion. The grassland type of northern China is classified by comprehensively considering the vegetation types and climate. From meadow steppe to typical steppes and desert steppes can reflect an aridity gradient from less to more arid. Site 9 located on the transitional area from typical steppe to desert steppe, which was classified to the desert steppe based on the vegetation types. We have conducted further analysis to examine the relationship between aridity and grazing effects (see Fig 1b, 2b), which was consistent with the results based on grassland type, grazing had stronger negative effect on biodiversity and multifunctionality in drier grasslands. Thus, our findings are solid. These detail information has been supplemented in lines 268-274.

(2) Table S1 should have detailed information for each site.

18. We have supplied the exclosure year of each site in Table S1, and more details about the study sites have been supplied in the method (see lines 259-281).

Reviewers' Comments:

Reviewer #1:

Remarks to the Author:

The authors have accommodated all my comments to my satisfaction. Very interesting and important contribution. Well done.

Jim Grace

Reviewer #2:

Remarks to the Author:

I have reviewed the revised version of paper titled "Experimental impacts of grazing on grassland biodiversity and function are explained by aridity" , and found that the authors have addressed all my concerns. I am very satisfied with their efforts to rewrite the paper.

Reviewer #1 (Remarks to the Author):

The authors have accommodated all my comments to my satisfaction. Very interesting and important contribution. Well done.

Jim Grace

1. Thank you for this positive feedback. We really appreciate this comment.

Reviewer #2 (Remarks to the Author):

I have reviewed the revised version of paper titled "Experimental impacts of grazing on grassland biodiversity and function are explained by aridity" , and found that the authors have addressed all my concerns. I am very satisfied with their efforts to rewrite the paper.

2. Thanks! We appreciate that this reviewer acknowledged the work we have done in previous revision to address all his/her comments.